# Increasing temperature of cooling granular gases

Nikolai V. Brilliantov[1], Arno Formella[2] & Thorsten Pöschel [3]

The kinetic energy of a force-free granular gas decays monotonously due to inelastic collisions of the particles. For a homogeneous granular gas of identical particles, the corresponding decay of granular temperature is quantified by Haff's law. Here, we report that for a granular gas of aggregating particles, the granular temperature does not necessarily decay but may even increase. Surprisingly, the increase of temperature is accompanied by the continuous loss of total gas energy. This stunning effect arises from a subtle interplay between decaying kinetic energy and gradual reduction of the number of degrees of freedom associated with the particles' dynamics. We derive a set of kinetic equations of Smoluchowski type for the concentrations of aggregates of different sizes and their energies. We find scaling solutions to these equations and a condition for the aggregation mechanism predicting growth of temperature. Numerical direct simulation Monte Carlo results confirm the theoretical predictions.

[1] Department of Mathematics, University of Leicester, University Road, Leicester LE1 7RH, UK. [2] Department of Computer Science, University of Vigo, Edificio Politécnico, 32004 Ourense, Spain. [3] Institute for Multiscale Simulation, Friedrich-Alexander-Universität Erlangen-Nürnberg, Nägelsbachstraße 49b, 91052 Erlangen, Germany. Correspondence and requests for materials should be addressed to T.P. (email: thorsten.poeschel@fau.de)

Dilute systems of inelastically colliding particles, also called granular gases, belong to the most intensively studied systems in the physics of granular matter[1,2]. Granular gases may be investigated using the tools of kinetic theory and hydrodynamics of molecular gases, with some modifications. These modifications are necessary to take into account the loss of mechanical energy due to particle collisions, quantified by the coefficient of restitution, defined as the ratio of post-collisional and pre-collisional relative normal velocity. Granular gases are inherently unstable as they tend to develop self-organized clusters[3,4] and other instabilities[5]. Although these instabilities inspired physicists to study granular gases, even the first stage in the evolution of a granular gas when it is homogeneous and isotropic reveals a rich phenomenology, which are fundamentally different from molecular gases. These may be exemplified by various types of correlations[4,6–9] and deviations from the Maxwell velocity distribution[10–16]. Based on an enormous amount of scientific literature in the past 25 years, by now the homogeneous state of granular gases is well understood[1,17–19].

An important feature of natural granular gases is the agglomeration of particles due to forces of different type. Such phenomena can be found, e.g., in form of soot-agglomeration in smoke gas[20], in aerosols[21,22], as well as in astrophysical systems such as dust in interplanetary space, in planetary rings consisting of aggregative ice particles[23] and protoplanetary disks[24–28]. While simple approaches assume that particle collision are always aggregative[28–32], more detailed models[27,28,33] take into account that only collisions at small impact rate lead to aggregation, while for larger impact velocity particles rebound or shatter[28,33]. Naturally, the mechanism of particle interactions determines conditions under which the particles aggregate, thus, a realistic model of aggregation should consider the physics of particle interactions, such as surface adhesion[27,28,33], or van der Waals interactions[34], electrostatics of charged or dipolar particles[35], or gravity[34].

Although it has been shown that the incorporation of aggregation changes the properties of granular gases considerably[28,30,31,33,35–38], by now the role of the physical aggregation mechanism was not considered. Moreover, while the size distribution of aggregates has been analyzed[27,28,33,35], a constant mean kinetic energy was assumed; a coupling between the evolution of the size distribution and that of the mean kinetic energy of the aggregates has not been investigated. It is however well known that the mean kinetic energy of the system is not constant but decreases with time[30,31,36–38], which strongly influences the kinetics of ballistic aggregation.

For an initially uniform gas of monomers, we analyze the evolution of the size distribution of aggregates and the average kinetic energy of these species for different realistic mechanisms of ballistic aggregation. From the Boltzmann equation for a multicomponent granular gas[1], we derive a set of coupled equations of Smoluchowski type for the concentration of aggregates of different sizes and the corresponding average kinetic energies. We obtain different regimes of the system's evolution characterized by either decreasing or increasing temperature. The increase of temperature is a surprising result given that all collisions of particles are dissipative.

## Results
**Kinetic theory.** We consider a dilute and uniform gas of granular particles and introduce the mass–velocity distribution function, $f_i \equiv f(m_i, \mathbf{v}_i, t)$, which is the concentration of particles of $i$ monomer masses at velocity $\mathbf{v}_i$ at time $t$. Its evolution obeys the

Boltzmann equation[28,33]:

$$\frac{\partial}{\partial t} f(m_k, \mathbf{v}_k, t) = I_k^{\mathrm{agg}} + I_k^{\mathrm{res}}, \tag{1}$$

where $I_k^{\mathrm{agg}}$ and $I_k^{\mathrm{res}}$ are, respectively, the collision integrals for the aggregative and restitutive collisions. Let us introduce the aggregation model and explain the structure of $I^{\mathrm{agg}}$ and $I^{\mathrm{res}}$. When non-adhesive granular particles of mass $m_i$ and $m_j$ collide, the energy of their relative motion, $E_{ij} = \frac{1}{2} m_i m_j \mathbf{v}_{ij}^2 / (m_i + m_j)$ with $\mathbf{v}_{ij} = \mathbf{v}_i - \mathbf{v}_j$, is reduced by the factor $\varepsilon^2$. The coefficient of restitution, $\varepsilon \leq 1$, thus, characterizes the inelastic nature of particle collisions[1]. For adhesive particles considered here, we assume that the particles agglomerate if the post-collisional relative kinetic energy is smaller than the energy needed to overcome the energy barrier $W_{ij}$ due to attractive forces, that is, $\varepsilon^2 E_{ij} \leq W_{ij}$. The energy, $W_{ij}$, depends on the size or the particles and the nature of the attractive forces. For instance, for adhesive forces, $W_{ij} = A \cdot (r_i r_j)^{4/3} (r_i + r_j)^{-4/3}$, where the constant $A$ depends on material parameters and $r_{i/j}$ are the particles' radii[39]. Let a particle of mass $m_i$ consist of $i$ monomers of mass $m_0$ and radius $r_0$. Then an aggregate of mass $m_i = m_0 i$ is of radius $r_i \sim i^{1/3}$. The general form

$$W_{ij} = a \frac{\left(i^{1/3} j^{1/3}\right)^{\lambda_1}}{\left(i^{1/3} + j^{1/3}\right)^{\lambda_2}}, \tag{2}$$

characterizes the dependence of $W_{ij}$ on the size of colliding particles, where $a$ is a constant of the dimension of energy. The choice of $\lambda_{1/2}$ describes a variety of attractive interactions. For example, $\lambda_1 = \lambda_2 = 4/3$ corresponds to the adhesive surface interactions introduced above; $\lambda_1 = 3$, $\lambda_2 = 1$ characterizes gravitational, or Coulomb interaction when the particles' charges scale as their masses. Similarly, $\lambda_1 = \lambda_2 = 3$ stands for dipole–dipole interactions[35], etc. Note that in the aggregation condition, we do not consider explicitly the particle rotation; we assume that its impact on the agglomeration kinetics may be effectively accounted for by the factor $a$ in Eq. (2).

For the case that particles aggregate, that is, $\varepsilon^2 E_{ij} \leq W_{ij}$, the corresponding collision integral reads[28,33]

$$\begin{aligned}
I_k^{\mathrm{agg}} = &\frac{1}{2} \sum_{i+j=k} \sigma_{ij}^2 \int d\mathbf{v}_i d\mathbf{v}_j d\mathbf{e}\, \Theta(-\mathbf{v}_{ij} \cdot \mathbf{e}) |\mathbf{v}_{ij} \cdot \mathbf{e}| \\
&f_i f_j \delta(m_k \mathbf{v}_k - m_i \mathbf{v}_i - m_j \mathbf{v}_j) \Theta_{ij} \\
&- \sum_j \sigma_{kj}^2 \int d\mathbf{v}_j d\mathbf{e}\, \Theta(-\mathbf{v}_{kj} \cdot \mathbf{e}) |\mathbf{v}_{kj} \cdot \mathbf{e}| f_k f_j \Theta_{kj}.
\end{aligned} \tag{3}$$

Here $\Theta_{ij} \equiv \Theta(W_{ij} - \varepsilon^2 E_{ij})$, with the Heaviside step-function, $\Theta(x)$, guarantees that the aggregation condition is fulfilled; $m_k = m_i + m_j$ and $m_k \mathbf{v}_k = \mathbf{v}_i m_i + m_j \mathbf{v}_j$, due to the mass and momentum conservation, $\sigma_{ij} = r_i + r_j$ and the inter-center unit vector $\mathbf{e}$ at the collision instant. The factors in the integrand in Eq. (3) have their usual meaning[1]: $\sigma_{ij}^2 |\mathbf{v}_{ij} \cdot \mathbf{e}|$ is the volume of the collision cylinder and $\Theta(-\mathbf{v}_{ij} \cdot \mathbf{e})$ selects only approaching particles. The first sum in the right-hand side of Eq. (3) refers to collisions, where a particle of size $k$ is formed from smaller particles of size $i$ and $j$, while the second sum describes the collisions of $k$-particles with all other aggregates. In case of restitutive collisions, the collision integral has its usual form[1],

$$\begin{aligned}
I_k^{\mathrm{res}} = &\sum_i \sigma_{ki}^2 \int d\mathbf{v}_i d\mathbf{e}\, \Theta(-\mathbf{v}_{ki} \cdot \mathbf{e}) |\mathbf{v}_{ki} \cdot \mathbf{e}| \left(\varepsilon^{-2} f_k'' \cdot f_i'' - f_k f_i\right) \\
&\Theta(\varepsilon^2 E_{ki} - W_{ki}),
\end{aligned} \tag{4}$$

with the additional factor $\Theta(\varepsilon^2 E_{ij} - W_{ij})$ to exclude an aggregation at the impact. Here, $f_{k/i}'' = f_{k/i}(\mathbf{v}_{k/i}'', t)$, with $\mathbf{v}_k''$

and $\mathbf{v}_i''$ being the velocities of the inverse collision, that results with the post-collision velocities $\mathbf{v}_k$ and $\mathbf{v}_i$[1].

Let us introduce the number density, $n_i$, of particles of size $i$, their partial temperature $T_i$, the total number density, $N$, and the average temperature, $T$[40]:

$$n_i = \int d\mathbf{v}_i f(m_i, \mathbf{v}_i) \qquad N = \sum_i n_i$$
$$3n_i T = \int d\mathbf{v}_i m_i \mathbf{v}_i^2 f(m_i, \mathbf{v}_i) \quad NT = \sum_i n_i T_i. \qquad (5)$$

We assume that the distribution function may be approximated by a Maxwellian,

$$f(m_i, \mathbf{v}_i, t) = \frac{n_i(t)}{\pi^{3/2} v_{0i}^3(t)} e^{-v_i^2/v_{0i}^2}, \qquad (6)$$

where $v_{0i}^2(t) = 2T_i(t)/m_i$ is the mean thermal velocity of the particles of size $i$. Multiplying Eq. (1) with unity and with $m_k \mathbf{v}_k^2$, using Eqs. (3) and (4) and integrating over $\mathbf{v}_k$, we obtain

$$\frac{d}{dt} n_k = \frac{1}{2} \sum_{i+j=k} C_{ij} n_i n_j - n_k \sum_j C_{ij} n_j$$
$$\frac{d}{dt} n_k \theta_k = \frac{1}{2} \sum_{i+j=k} B_{ij} \frac{n_i n_j \theta_i \theta_j}{\theta_i + \theta_j} - \sum_j D_{kj} \frac{n_k n_j \theta_k \theta_j}{\theta_i + \theta_j}, \qquad (7)$$

where $\theta_i = T_i/m_i$ and the coefficients $C_{ij}$, $B_{ij}$, and $D_{ij}$ depend on $\sigma_{ij}$, $W_{ij}$, $m_i$, $m_j$, and $\theta_i$, $\theta_j$, see "Methods" section for details. The set of Eq. (7) for the concentrations of species (i.e., of the agglomerates of size $k$) and their average kinetic energy are the extended Smoluchowski-type equations: They describe the aggregation processes when different species have time-dependent individual temperatures. For non-aggregative particles, that is for $W_{ij} = 0$, we obtain $C_{ij} = B_{ij} = 0$ and $D_{ij} = \xi_{ij}(\theta_i + \theta_j)/(n_j \theta_j)$, where $\xi_{ij}$ are the cooling coefficients[41,42]. The set of Eq. (7) then reduces to $dT_k/dt = -T_k \sum_i \xi_{ki}$ in agreement with refs.[41,42].

From numerical simulations (see below) we find that the partial temperatures evolve as $T_i(t) = \phi_i T(t)$, where the constants $\phi_i$ weakly depend on $i$. For the qualitative analysis we assume $T_i(t) \approx T(t)$, where $T(t)$ is the characteristic temperature of the system. Then Eq. (7) reduce to

$$\frac{d}{dt} n_k = \frac{1}{2} \sum_{i+j=k} C_{ij}(T) n_i n_j - n_k \sum_j C_{ij}(T) n_j \qquad (8)$$

$$\frac{d}{dt} NT = -\sum_{i,j} P_{ij}(T) n_i n_j, \qquad (9)$$

with

$$C_{ij} = \nu_{ij} \left[ 1 - \left( 1 + \tilde{W}_{ij}/T \right) e^{-\tilde{W}_{ij}/T} \right]$$
$$\nu_{ij} = 2\sigma_{ij}^2 \sqrt{2\pi T/\mu_{ij}}$$
$$P_{ij} = \frac{2}{3} \nu_{ij} T \left( 1 - G_{ij} + \frac{1}{2}(1 - \varepsilon^2) G_{ij} \right) \qquad (10)$$
$$G_{ij} = \left( 1 + \tilde{W}_{ij}/T + \frac{1}{2} \tilde{W}_{ij}^2/T^2 \right) e^{-\tilde{W}_{ij}/T},$$

and we abbreviate $\tilde{W}_{ij} \equiv W_{ij}/\varepsilon^2$ and $\mu_{ij} = m_i m_j/(m_i + m_j)$. Below, we will consider some limiting cases.

**Hot gas limit.** For $(a/T) \ll 1$, that is, $\tilde{W}_{ij}/T \ll 1$ for all particles, the aggregation barrier is much smaller than the average kinetic energy of the particles. From Eq. (10) then follows $C_{ij} \sim (a/T)^2$ and $P_{ij}$ contains two terms, one $\sim (1 - \varepsilon^2)$ and the other one $\sim (a/T)^3$. This entails different regimes of gas behavior:

Non-aggregative cooling occurs when $(a/T) \to 0$, thus, $C_{ij} \approx 0$, Eq. (8) yields $\dot{n}_k = \dot{N} = 0$, which implies the equation

$$\dot{T} = -\zeta T^{3/2}; \; \zeta = \sum_{i,j} \frac{2}{3} \sqrt{\frac{2\pi}{\mu_{ij}}} (1 - \varepsilon^2) \sigma_{ij}^2 \frac{n_i n_j}{N}, \qquad (11)$$

with Haff's solution[43] for a non-aggregative cooling gas, $T \sim (1 + t/\tau_0)^{-2}$, where $\tau_0 = \zeta \sqrt{T(0)}$.

Partial aggregation with cooling takes place for smaller temperature. In this case, we take into account aggregation but disregard the cooling due to aggregation. Then $C_{ij} \sim (a/T)^2$ and $P_{ij} \sim (1 - \varepsilon^2)$; in more detail, $C_{ij} = b_1 T^{-3/2} \tilde{C}_{ij}$ and $P_{ij} = b_2 T^{3/2} \tilde{P}_{ij}$, with the dimensionless kernels

$$\tilde{C}_{ij} = (ij)^{2\lambda_1/3 - 1/2} (i+j)^{1/2} \left( i^{1/3} + j^{1/3} \right)^{2 - 2\lambda_2} \qquad (12)$$

$$\tilde{P}_{ij} = (ij)^{-1/2} (i+j)^{1/2} \left( i^{1/3} + j^{1/3} \right)^2, \qquad (13)$$

and constants $b_1 = k_0 (a/\varepsilon^2)^2$, $b_2 = (2/3) k_0 (1 - \varepsilon^2)$ and $k_0 = r_0^2 \sqrt{2\pi/m_0}$. The kernels are homogeneous functions of $i$ and $j$:

$$\tilde{C}_{si\,sj} = s^{\mu_c} \tilde{C}_{ij}; \; \tilde{P}_{si\,sj} = s^{\mu_p} \tilde{P}_{ij}, \qquad (14)$$

where $\mu_p = 1/6$ and $\mu_c = 2(2\lambda_1 - \lambda_2)/3 + 1/6 = (2/3)\Lambda + 1/6$. Here we introduce $\Lambda = 2\lambda_1 - \lambda_2$, where $\Lambda/3$ is the homogeneity degree of the attraction barrier (2).

For large time, $t$, Smoluchowski equations with homogeneous kernels give rise to scaling solutions[44–46], therefore, we seek solution to Eqs. (8) and (9) in the scaling form

$$n_k \simeq t^{-2z} \Phi(k/t^z); \; T \sim t^{-\beta}. \qquad (15)$$

From Eq. (15) follows $N(t) \sim t^{-z}$. Substituting Eq. (15) into Eqs. (8) and (9), using the homogeneity of kernels and approximating the discrete variables by continuous variables, $\tilde{C}_{ij} \to \tilde{C}(i, j)$, one obtains the equation for the scaling function $\Phi(x)$ (see more details in "Methods" section),

$$w[x\Phi' + 2\Phi] = \Phi(x) \int_0^\infty dy\, \tilde{C}(x, y) \Phi(y)$$
$$- \frac{1}{2} \int_0^x dy\, \tilde{C}(x-y) \Phi(x-y) \Phi(y), \qquad (16)$$

where $w$ is the separation constant[45,46]. Equation (16) is to be supplemented by the relations for the exponents $z$ and $\beta$, which follow from the requirement that the power of $t$ should be the same on both sides of Eqs. (8) and (9):

$$z(1 - \mu_c) = 1 + 3\beta/2; \; z\left(1 - \mu_p\right) = 1 - \beta/2. \qquad (17)$$

Solving for $z$ and $\beta$ and using the expressions for $\mu_c$ and $\mu_p$, we obtain for this regime,

$$z = \frac{2}{5 - \Lambda}; \; \beta = \frac{\Lambda}{5 - \Lambda}. \qquad (18)$$

Note that the scaling analysis is valid if $z > 0$, therefore, $5 - \Lambda > 0$. If this condition is violated a more subtle analysis is needed; this may indicate a gelation[44–46] or run-away growth[35].

Aggregation with temperature growth may take place for a molecular gas ($\varepsilon = 1$) or nearly elastic granular gas. In this case, $(1 - \varepsilon^2) \ll (a/T)^3$ and we obtain, $C_{ij} = b_1 T^{-3/2} \tilde{C}_{ij}$, with $\tilde{C}_{ij}$ given by Eq. (13), while $P_{ij} \sim (a/T)^3$. It may also be written as

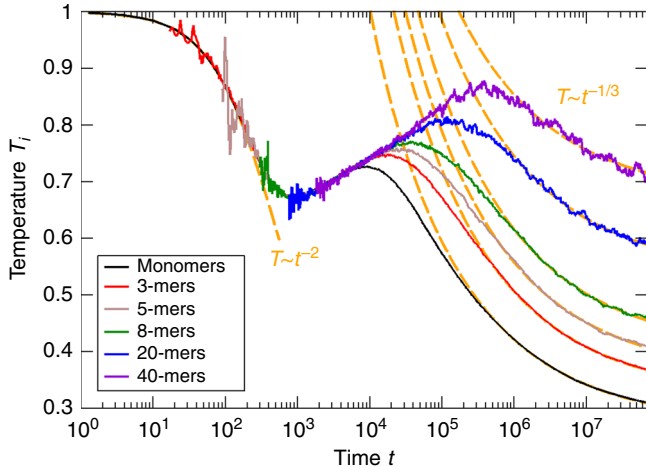

**Fig. 1** Partial temperatures of *i*-mers. Evolution of the partial temperatures, $T_i$, of *i*-mers for a granular gas of $10^7$ particles for $\lambda_1 = \lambda_2 = 4/3$, $a = 0.1$, and $\varepsilon = 0.99$. Initially, the gas of monomers has the temperature, $T_1(0) = 1$. The dashed lines show the limiting cases of a hot gas (regime of non-aggregative cooling) when the temperature follows Haff's law, $T \sim t^{-2}$, and cold gas when almost all collisions are aggregative and $T \sim t^{-1/3}$. Both cases are in agreement with the theory, see Eqs. (11) and (25)

$P_{ij} = b_3 T^{-3/2} \tilde{P}_{ij}$, where $b_3 = (2/9)k_0(a/\varepsilon^2)^3$ and $\tilde{P}_{ij}$ reads

$$\tilde{P}_{ij} = (ij)^{\lambda_1 - 1/2}(i+j)^{1/2}\left(i^{1/3} + j^{1/3}\right)^{2-3\lambda_2}, \quad (19)$$

that is, $\tilde{P}_{ij}$ is a homogeneous function with the homogeneity constant $\mu_{p,1} = \Lambda + 1/6$.

The scaling analysis given above yields again Eq. (16) for the scaling function and the relations for the exponents $z$ and $\beta$:

$$z(1 - \mu_c) = 1 + 3\beta/2; \quad z\left(1 - \mu_{p,1}\right) = 1 + 5\beta/2. \quad (20)$$

From Eq. (20), we find

$$z = \frac{6}{5 - \Lambda}; \quad \beta = -\frac{2\Lambda}{5 - \Lambda}. \quad (21)$$

The above result is surprising: If $\Lambda > 0$, which corresponds to the increase of the attractive barrier with aggregates' size, the exponent $\beta$ is negative, that is, $T \sim t^{|\beta|}$, and the characteristic temperature of an aggregating gas increases! This effect may be understood as follows: For an elastic or nearly elastic gas, the aggregation conditions (defined by the constants $\lambda_1$ and $\lambda_2$) may lead to a regime, when the total number of particles $N$ decays faster than the total energy of the system. In this regime, the increase of the gas temperature occurs during its cooling, hence it behaves like a system with a negative-specific heat[47,48], see the discussion below.

**Cold gas limit.** Complete aggregation with cooling occurs in the limit $(a/T) \gg 1$, such that $W_{ij}/T \gg 1$ for all species. The aggregation barrier is much larger than the average kinetic energy of the particles, therefore, almost all the collisions lead to aggregation. Then from Eq. (10) follows, $C_{ij} = \nu_{ij} = 2k_0 T^{1/2}\tilde{C}_{ij}$ and $P_{ij} = (2/3)T\nu_{ij} = (4/3)k_0 T^{3/2}\tilde{P}_{ij}$ where

$$\tilde{C}_{ij} = \tilde{P}_{ij} = (ij)^{-1/2}(i+j)^{1/2}\left(i^{1/3} + j^{1/3}\right)^2, \quad (22)$$

that is, both kernels $\tilde{C}_{ij}$ and $\tilde{P}_{ij}$ are homogeneous with the homogeneity constant $\mu_{c,1} = \mu_{p,2} = 1/6$. From Eq. (22) follows $\sum_{i,j} P_{ij}n_i n_j = (2/3)T\sum_{i,j} C_{ij}n_i n_j$, then Eqs. (8) and (9) entail the

equation:

$$T\frac{dN}{dt} + N\frac{dT}{dt} = \frac{4}{3}T\frac{dN}{dt}. \quad (23)$$

The scaling analysis of Eqs. (8) and (23) yields again Eq. (16) for $\Phi(x)$ and the relations

$$z\left(1 - \mu_{p,2}\right) = 1 - \beta/2; \quad z = 3\beta, \quad (24)$$

which finally give the exponents,

$$z = 1; \quad \beta = 1/3. \quad (25)$$

This is the aggregation regime of ordinary Smoluchowski type, when all collisions lead to agglomeration of the particles.

**Numerical confirmation.** To check the predictions of the previous section, we performed large-scale Monte Carlo simulations. The details of the numerical method are given in "Methods" section below. Figure 1 shows the evolution of the partial temperatures, $T_i$, of *i*-mers.

While for an ordinary granular gas, temperature would monotonously decay due to Haff's law, $T(t) \sim t^{-2}$, in case of a gas of aggregative particles, temperature evolves non-monotonously by revealing a number of different regimes. We start with a homogeneous gas of monomers at temperature $T_1(0) = 1$. Then initially, the condition $(a/T) \ll 1$ is satisfied and the gas cools according to Haff's law. This corresponds to the regime of the non-aggregative cooling for a hot gas described by Eq. (11). In the course of time, temperature drops down, the aggregation becomes non-negligible such that the evolution of temperature deviates from Haff's law; the system now evolves according to the regime of partial aggregation with cooling. With further decrease of temperature and increase of the ratio $(a/T)$, aggregation dominates over dissipative loss for the evolution of temperature. This corresponds to the regime of aggregation with temperature growth, where the characteristic temperature of the system increases as $T \sim t^{|\beta|}$, see Eq. (21). Eventually, the latter regime terminates because the condition $(1 - \varepsilon^2) \ll (W_{ij}/T)^3$ needed for this regime is violated with increasing temperature. The larger the size $i$ of a particle, the larger the value of $W_{ij}$ and hence the large temperature $T_i$, when the change of the regimes takes place; this is clearly visible in Fig. 1. In the final state of its evolution, the gas approaches the limit of a cold gas, when almost all collisions are aggregative. In this regime, the temperature decays with time as $T \sim t^{-1/3}$, in agreement with the theoretical prediction, Eq. (25).

Figure 2 shows the regime of aggregation with temperature growth in more detail. In agreement with the theoretical prediction, Eq. (21), the increase of temperature in this regime becomes steeper with increasing value of $\Lambda$, which is proportional to the homogeneity degree $\Lambda/3$ of the aggregation barrier.

It is also interesting to estimate the onset time, $t_h$, when the granular temperature starts to increase. This regime occurs in a hot granular gas when $(1 - \varepsilon^2) \ll (a/T)^3 \ll 1$. If initially $(a/T)^3$ is much smaller than $(1 - \varepsilon^2)$, the regime of non-aggregative cooling takes place and temperature decreases according to Haff's law, $T(t) \sim 1/(t/\tau_0)^2$. Here, $\tau_0^{-1} = (1 - \varepsilon^2)4\eta/\sigma\sqrt{T(0)/\pi m} = (1 - \varepsilon^2)\tau_c^{-1}/6$ with $\eta$ being the packing fraction of the gas (see the "Methods" section) and $\tau_c$ is the initial mean collision time. In the course of time, temperature decrease so that $(a/T)^3$ grows. For the order of magnitude estimate of the onset time $t_h$ one can use the condition $(1 - \varepsilon^2) \approx (a/T(t_h))^3$. For the parameters $\varepsilon = 0.99$, $a = 0.1$, $T(0) = 1$, $\sigma = 1$, $m = 1$, and $\eta = 0.05$ used in the simulations depicted in Figs. 1 and 2, we obtain $t_h \simeq 500$. This qualitatively

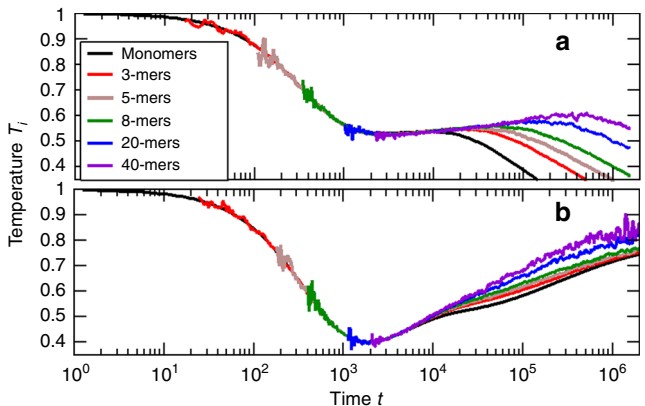

**Fig. 2** Evolution of temperature for different aggregation mechanisms. The rate of temperature growth in the regime of increasing temperature depends on the aggregation mechanism, quantified by the parameter $\Lambda$. This parameter characterizes the dependence of the aggregation barrier on the agglomerate size. **a** $N = 10^7$, $\varepsilon = 0.99$, $a = 0.1$, $\Lambda = 0.4$, thus $\beta = -0.173$, see Eq. (21); **b** same but $\Lambda = 1.6$, thus $\beta = -0.941$. With increasing value of $\Lambda$, the increase of temperature becomes steeper, in agreement with the theoretical predictions, $T \sim t^{|\beta|}$, of Eq. (21)

agrees with the numerical result, of $t_h \approx 1000$, which corresponds to approximately 600 collisions with the duration of $\tau_c \simeq 1.5$, see Figs. 1 and 2.

**Conditions for experimental observations**. We estimate the range of parameters where the surprising phenomenon of increasing temperature of a cooling granular gas may be observed. As it follows from the above analysis, this is expected for a hot gas, when $(1 - \varepsilon^2) \ll (a/T)^3 \ll 1$. Note, however, that while the condition $(1 - \varepsilon^2) \ll (a/T)^3$ allows a simplified analytical treatment, a milder condition $(1 - \varepsilon^2) < (a/T)^3$ ensures the increase of temperature as well.

Let us consider the case of adhesive interactions associated with the van der Waals forces[34,49]. In this case, the adhesion energy $W_{ij}$ defined in Eq. (2) reads[39],

$$W_{ij} = q_0 \left( \pi^5 \gamma^5 R_{ij}^4 D^2 \right)^{1/3},$$ (26)

where $\gamma$ is the adhesion coefficient, which is twice the surface free energy per unit area of a solid in vacuum, $R_{ij} = r_i r_j/(r_i + r_j)$, $D = \frac{3}{2}(1 - \nu^2)/Y$, with $\nu$ and $Y$ being, respectively, Poisson ratio and Young modulus, and $q_0 = 1.457$ is a numerical constant. With $r_i = r_0 i^{1/3}$, where $r_0$ is the monomer radius, we estimate,

$$W_{ij} = a \frac{\left( i^{1/3} j^{1/3} \right)^{4/3}}{\left( i^{1/3} + j^{1/3} \right)^{4/3}},$$

which implies the homogeneity exponents $\lambda_1 = \lambda_2 = 4/3$ and the constant $a$ in Eq. (2):

$$a^3 = q_0^3 \pi^5 \gamma^5 r_0^4 D^2.$$ (27)

Since the homogeneity degree of $W_{ij}$ is positive, $\Lambda/3 = (2\lambda_1 - \lambda_2)/3 = 4/9 > 0$, the regime of aggregation with temperature growth is possible.

Using the data available in the literature for the restitution coefficient[50,51] and material parameters ($\gamma$, $Y$, $\nu$, and the material density $\rho$) we obtain, that for a granular gas of very hard ceramic particles ($Y = 370 \times 10^9$ GPa) of diameter $\sigma = 1$ mm moving at the characteristic velocity $v_0 = 0.16$ ms$^{-1}$ ($T = mv_0^2/3$), the value of $1 - \varepsilon^2 \simeq 0.073$ is indeed significantly smaller than

$(a/T)^3 \simeq 0.32$. For the opposite case of rather soft acrylic particles ($Y = 3 \times 10^9$ GPa), similar estimates indicate that such regime may be realized for a granular gas of particles of the same size but for the characteristic velocity $v_0 = 0.27$ ms$^{-1}$; in this case, $1 - \varepsilon^2 \simeq 0.15$ and $(a/T)^3 \simeq 0.48$. Hence, we conclude that the regime of increasing temperature of a cooling granular gas may be observed for many realistic systems.

Let us estimate the onset time $t_h$ for real systems discussed above, which are prepared at the hot gas conditions, $(a/T)^3 \ll (1 - \varepsilon^2)$ and then freely evolve. Suppose that initially $(a/T(0))^3 = 0.001$. Then using the estimates for $t_h$ outlined above, we obtain that $t_h = 17$ s for the first system of (hard) ceramic particles of diameter $\sigma = 1$ mm; the gas is to be prepared with the packing fraction $\eta = 0.05$ and initial characteristic velocity $v_0(0) = 3$ ms$^{-1}$. Similarly, we obtain $t_h = 72$ s for the second system of (soft) acrylic particles of the same diameter, prepared at the same packing fraction with the characteristic velocity $v_0(0) = 6$ ms$^{-1}$.

Experimental investigations of cooling granular gases are not simple since to assure force-free conditions, the action of gravity has to be suppressed. Such experiments have been performed, either under true microgravity conditions aboard the space station[52], in parabolic flights[53,54], or in sounding rockets[55,56], or by means of magnetic levitation[57]. By now, however, the focus of these experiments was to check experimentally the cooling law theoretically predicted by Haff and others[43]. Hence, the experiments have been designed to assure purely repulsive interaction of the particles, that is, to keep attractive forces as small as possible. This might be the reason why the temporary increase of temperature reported in our paper was not mentioned in any of these studies. As it follows from the estimates given above, to ensure the emergence of the regime of interest, one needs to prepare a system at particular conditions.

**Analogy with systems of negative-specific heat**. It is worth to note that the behavior of an aggregating granular gas with increasing temperature may be interpreted as a manifestation of a negative-specific heat: The energy of the system decreases in dissipative collisions, while the granular temperature grows, that is, the smaller the energy of the gas, the larger its temperature. The negative-specific heat characterizes equilibrium systems with long-range (e.g., gravitational) interactions, see e.g., refs. [47,48]. Such analogy is very interesting and tempting. Still however, the direct application of this concept to non-equilibrium systems, addressed in our study, is to be justified by further analysis. In particular, one needs to investigate the opposite process, when temperature decreases with increasing system energy. This may be possible for an extended model, which includes shattering collisions and mechanisms of energy input into the system. Then, if the number of particles, emerging in shattering impacts, increases faster than energy of the system, thanks to the energy input, the temperature of the gas would decrease, manifesting a negative heat capacity. To apply the concept of negative heat capacity to aggregating gases, a detailed analysis of the thermodynamic additivity[47,48] in their cooling and heating states will also be needed. We leave these fascinating problems for future studies.

**Discussion**
By means of kinetic theory, we show that a force-free gas of aggregative particles behaves fundamentally different than an ordinary granular gas of purely repulsive particles. In both cases, the kinetic energy decreases monotonously due to dissipative collisions of the particles. However, while the temperature of ordinary granular gases decreases monotonously as well, according to Haff's law, $T(t) \sim t^{-2}$, the evolution of granular gases

of aggregating particles demonstrates subsequent regimes of very different nature. These regimes are characterized by either decreasing or increasing temperature. The astonishing fact that the temperature of a gas of particles, which dissipate energy increases, may be understood as follows: although the total energy of the system decreases, the total number of particles diminishes faster due to agglomeration, yielding a boost of energy per particle. This surprising effect is possible if the aggregation barrier increases with the size of the aggregates (i.e., $\Lambda > 0$), and, as we have shown above, may be observed for many realistic systems. Interestingly, the increase of temperature with decaying energy corresponds to a negative-specific heat for equilibrium systems[47,48]; the application of this concept to non-equilibrium systems requires however further analysis.

Technically, we derived an extended set of Smoluchowski equations: equations for concentrations of different species, $n_k$, which correspond to standard Smoluchowski equations and equations for the partial temperatures of the species, $T_k$. Using the approximation $T_k(t) \approx T(t)$, we elaborate a scaling theory and reveal several regimes of temperature evolution, including the surprising regime of increasing temperature of a cooling gas. Numerical Monte Carlo simulations confirm the predictions of our theory.

## Methods

**Kinetic kernels of extended Smoluchowski equations.** To derive the first set of rate Eq. (7) for $n_k(t)$, we integrate the Boltzmann equation, Eq. (1), over $\mathbf{v}_k$. Since $n_k = \int d\mathbf{v}_k f_k(\mathbf{v}_k, t)$, the left-hand side of the Boltzmann equation turns then into $dn_k/dt$ and gives the rate of change of the concentrations $n_k$. The right-hand side gives the contributions to $dn_k/dt$ from different parts of the collision integral. Since the restitutive (non-aggregating) collisions preserve the number of particles, we find

$$\int d\mathbf{v}_k I_k^{\text{res}} = 0. \tag{28}$$

To compute $\int d\mathbf{v}_k I_k^{\text{agg}}$, we assume that the velocity distribution function may be approximated by a Maxwellian, Eq. (6). For a non-aggregating granular gas, the velocity distribution function deviates from Maxwellian; the deviation may be expressed in terms of Sonine polynomials expansion[10–14,16]. These deviations are however small and, as it has been shown in refs. [41,42,58], may be neglected when the cooling coefficients $\xi_{ij}$ are computed. Therefore, the approximation (6) is justified.

The collision integral for aggregative collisions, given by Eq. (3) may be written as a sum of two parts,

$$I_k^{\text{agg}}(\mathbf{v}_k) \equiv I_k^{\text{agg},1} - I_k^{\text{agg},2}. \tag{29}$$

First, we compute $\int d\mathbf{v}_k I_k^{\text{agg},2}$,

$$\begin{aligned}
\int d\mathbf{v}_k I_k^{\text{agg},2} &= \sum_j \sigma_{kj}^2 \int d\mathbf{v}_k \int d\mathbf{v}_j \int d\mathbf{e}\, \Theta(-\mathbf{v}_{kj} \cdot \mathbf{e})|\mathbf{v}_{kj} \cdot \mathbf{e}| \\
&\quad f_k(\mathbf{v}_k) f_j(\mathbf{v}_j) \Theta(W_{ij} - \varepsilon^2 E_{ij}) \\
&= \sum_j \frac{\sigma_{kj}^2 n_k n_j}{\pi^3 v_{0,k}^3 v_{0,j}^3} \int d\mathbf{v}_k d\mathbf{v}_j d\mathbf{e}\, \Theta(-\mathbf{v}_{kj} \cdot \mathbf{e})|\mathbf{v}_{kj} \cdot \mathbf{e}| \\
&\quad e^{-v_k^2/v_{0,k}^2 - v_j^2/v_{0,j}^2} \Theta\left(W_{ij} - \varepsilon^2 \frac{\mu_{kj} v_{kj}^2}{2}\right),
\end{aligned} \tag{30}$$

where we use Eq. (6) for the velocity distribution function. The integrals in the above equation are Gaussian and hence may be straightforwardly calculated. We perform this calculation for a particular pair, $k$ and $j$. With the substitute

$$\mathbf{v}_k = \mathbf{u} + \mathbf{w}\left(\frac{\mu_{kj}}{m_k} - p_{kj}\right); \quad \mathbf{v}_j = \mathbf{u} - \mathbf{w}\left(\frac{\mu_{kj}}{m_j} + p_{kj}\right), \tag{31}$$

where

$$p_{kj} = \mu_{kj} \frac{\left(m_k v_{0,k}^2\right)^{-1} - \left(m_j v_{0,j}^2\right)^{-1}}{v_{0,k}^{-2} + v_{0,j}^{-2}}, \tag{32}$$

the above integral with respect to $\mathbf{v}_k$, $\mathbf{v}_j$, and $\mathbf{e}$ may be generally written as

$$\int d\mathbf{u}\, d\mathbf{w}\, d\mathbf{e}\, \Theta(-\mathbf{w} \cdot \mathbf{e})|\mathbf{w} \cdot \mathbf{e}| u^\alpha w^\beta |\mathbf{w} \cdot \mathbf{e}|^\gamma e^{-a_{kj} u^2 - b_{kj} w^2} \Theta\left(\frac{1}{\varepsilon^2} W_{ij} - \frac{1}{2}\mu_{kj} w^2\right), \tag{33}$$

where $a_{kj} = v_{0,k}^{-2} + v_{0,j}^{-2}$, $b_{kj} = \left(v_{0,k}^2 + v_{0,j}^2\right)^{-1}$, and $\alpha = \beta = \gamma = 0$, for the particular

case of the integrals in Eq. (30). We also take into account that the Jacobian of transformation from $(\mathbf{v}_k, \mathbf{v}_j)$ to $(\mathbf{u}, \mathbf{w})$ is equal to unity. Integration over $\mathbf{u}$ gives $(\pi/a_{kj})^{3/2}$. Integration over the unit vector $\mathbf{e}$ gives $4\pi$ and we are left with the integral over $\mathbf{w}$. Integration over directions of the vector $\mathbf{w}$ gives $\pi$, so finally we need to compute the remaining integral:

$$h_{kj} = \int_0^{\sqrt{\frac{2W_{kj}}{\varepsilon^2 \mu_{kj}}}} dw\, w^3 e^{-b_{kj} w^2} = \frac{1}{2b_{kj}^2}\left[1 - e^{\frac{2b_{kj} W_{kj}}{\varepsilon^2 \mu_{kj}}}\left(1 + \frac{2b_{kj} W_{kj}}{\varepsilon^2 \mu_{kj}}\right)\right]. \tag{34}$$

As the result we obtain:

$$\int d\mathbf{v}_k I_k^{\text{agg},2} = \sum_j \frac{\sigma_{kj}^2 n_k n_j}{\pi^3 v_{0,k}^3 v_{0,j}^3} 4\pi^2 \left(\frac{\pi}{a_{kj}}\right)^{3/2} h_{kj}. \tag{35}$$

Similarly, one can find the integral in Eq. (33) for other values of $\alpha \neq 0$, $\beta \neq 0$, and $\gamma \neq 0$.

Let us now calculate $\int d\mathbf{v}_k I_k^{\text{agg},1}$. First, we notice that $\int d\mathbf{v}_k \delta(m_k \mathbf{v}_k - m_i \mathbf{v}_i - m_j \mathbf{v}_j) = 1$, since the other part of the integrand does not depend on $\mathbf{v}_k$. Then the remaining integration is exactly the same as for $I_k^{\text{agg},2}$, which have been already performed, therefore we find:

$$\int d\mathbf{v}_k I_k^{\text{agg},1} = \frac{1}{2}\sum_{i+j=k} \frac{\sigma_{ij}^2 n_i n_j}{\pi^3 v_{0,i}^3 v_{0,j}^3} 4\pi^2 \left(\frac{\pi}{a_{ij}}\right)^{3/2} h_{ij}. \tag{36}$$

Hence, after some algebra, we obtain,

$$\frac{d}{dt} n_k = \frac{1}{2}\sum_{i+j=k} C_{ij} n_i n_j - n_k \sum_j C_{ij} n_j, \tag{37}$$

where

$$\begin{aligned}
C_{ij} &= 2\sqrt{2\pi}\sigma_{ij}^2 \sqrt{\theta_i + \theta_j}(1 - F_{ij}) \\
F_{ij} &= (1 + Q_{ij}) e^{-Q_{ij}} \\
Q_{ij} &= \frac{W_{ij}}{\varepsilon^2 \mu_{ij}(\theta_i + \theta_j)} \\
\theta &= T_i/m_i.
\end{aligned} \tag{38}$$

To derive the second set of rate equations in the system, Eq. (7), for $n_k(t)\theta_k(t)$, we multiply the Boltzmann equation, Eq. (3), with $m_k v_k^2/2$ and integrate over $\mathbf{v}_k$. Since $3n_k T_k = \int d\mathbf{v}_k m_k v_k^2 f_k(\mathbf{v}_k, t)$, the left-hand side of the equation turns into $3m_k d/dt(n_k \theta_k)$. In the right-hand side of this equation we again encounter the Gaussian integrals, as in Eq. (33), but with different values of $\alpha$, $\beta$, and $\gamma$. These integrals may be computed exactly in the same way as it has been shown above for the particular case, $\alpha = \beta = \gamma = 0$. Finally, we arrive at the set of equations for $\theta_k = T_k/m_k$,

$$\frac{d}{dt} n_k \theta_k = \frac{1}{2}\sum_{i+j=k} B_{ij} \frac{n_i n_j \theta_i \theta_j}{\theta_i + \theta_j} - \sum_j D_{kj} \frac{n_k n_j \theta_k \theta_j}{\theta_k + \theta_j} \tag{39}$$

with the coefficients:

$$\begin{aligned}
B_{ij} &= 2\sqrt{2\pi}\sigma_{ij}^2 \sqrt{\theta_i + \theta_j} \\
&\quad \times \left[1 - F_{ij} + \frac{4}{3}\frac{(\theta_i \Delta_{ij} - \theta_j \Delta_{ji})^2}{\theta_i \theta_j}(1 - G_{ij})\right] \\
D_{ij} &= 2\sqrt{2\pi}\sigma_{ij}^2 \sqrt{\theta_i + \theta_j} \\
&\quad \times \left[1 - F_{ij} + \frac{4}{3}\frac{\theta_i}{\theta_j}(1 - G_{ij}) + \frac{4}{3}\frac{\mu_{ij}}{m_i \theta_j}(\theta_i + \theta_j)\right. \\
&\quad \times \left. (1 + \varepsilon)\left(1 - \frac{1}{2}(1 + \varepsilon)\frac{\mu_{ij}}{m_i \theta_i}(\theta_i + \theta_j)\right) G_{ij}\right] \\
G_{ij} &= e^{-Q_{ij}}\left(1 + Q_{ij} + \frac{1}{2}Q_{ij}^2\right)
\end{aligned} \tag{40}$$

where $Q_{ij}$ has been defined above and $\Delta_{ij} = m_i/(m_i + m_j)$.

For $W_{ij} = 0$ we obtain $C_{ij} = B_{ij} = 0$ and

$$\begin{aligned}
D_{ij} &= \frac{8}{3}\sqrt{2\pi}\sigma_{ij}^2 \frac{\mu_{ij}}{m_i \theta_j}(\theta_i + \theta_j)^{3/2}(1 + \varepsilon) \\
&\quad \left[1 - \frac{1}{2}(1 + \varepsilon)\frac{\mu_{ij}}{m_i \theta_i}(\theta_i + \theta_j)\right] = \xi_{ij}\frac{\theta_i + \theta_j}{n_j \theta_j},
\end{aligned} \tag{41}$$

where $\xi_{ij}$ are the cooling coefficients for a non-aggregating granular mixture[41,42,58]. Then Eq. (39) takes the form,

$$\frac{dT_k}{dt} = -T_k \sum_j \frac{D_{kj} n_j \theta_j}{\theta_k + \theta_j} = -T_k \sum_j \xi_{kj}, \tag{42}$$

in agreement with refs. [41,42,58].

If $T_i = T$ for all $i$, we find $Q_{ij} = W_{ij}/(\varepsilon^2 T) = \tilde{W}_{ij}/T$ and arrive at Eq. (11) for $C_{ij}$. The coefficients $P_{ij}$ of Eq. (10) may be obtained noticing that

$$P_{ij} = \frac{1}{2}\left[ m_i D_{ij} + m_j D_{ji} - (m_i + m_j) B_{ij} \right]. \tag{43}$$

Using then Eqs. (40) for $B_{ij}$ and $D_{ij}$, we arrive for $T_i = T$ at $P_{ij}$, given by Eq. (10).

**Scaling analysis of the extended Smoluchowski equations.** Generally, it is not possible to solve analytically the infinite set of Eqs. (8) and (9), however, one can find a scaling solution, which reflects the most prominent features of the exact solution. To understand the nature of the scaling approach, we first consider a non-rigorous, qualitative analysis and then focus on its rigorous counterpart. Exploiting the basic scaling hypothesis[45,46] we assume that for large time and large $k$ a scaling solution to Eqs. (8) and (9) is attained. Hence, we seek the solution to these equations in the form

$$n_k \simeq s^{-2}\Phi(k/s), \tag{44}$$

where $s(t)$ is the characteristic aggregate size. It may be defined through the second moment as $s = M_2/M_1$, where $M_\alpha = \sum_{i=k}^{\infty} k^\alpha n_k$ and $M_1 = \mathrm{const.}$ is the total mass which is conserved. Multiplying Eq. (8) with $k^2$ we obtain

$$\frac{ds}{dt} = \frac{1}{M_1}\sum_{i,j} C_{ij} n_i n_j ij \sim C_{ss}\left(\sum_i n_i i\right)^2 \sim T^{3/2} s^{\mu_c}, \tag{45}$$

where we use the condition of the mass conservation along with the result of the main text for the dimensionless kernel:

$$C_{ss} \sim T^{3/2}\tilde{C}_{ss} \sim T^{3/2}s^{\mu_c}\tilde{C}_{11} \sim T^{3/2}s^{\mu_c}. \tag{46}$$

Taking into account that the total number of aggregates and its average size are related as $N \sim M_1/s \sim 1/s$, we recast Eq. (9) into the form

$$\frac{d}{dt}\left(\frac{T}{s}\right) \sim -\sum_{i,j} P_{ij} n_i n_j \sim -P_{ss}\left(\sum_i n_i\right)^2 \sim -T^{3/2}s^{\mu_p}N^2 \sim -T^{3/2}s^{\mu_p}s^{-2}, \tag{47}$$

where we again use the relation for the dimensionless kernel: $P_{ss} \sim T^{3/2}\tilde{P}_{ss} \sim T^{3/2}s^{\mu_p}\tilde{P}_{11} \sim T^{3/2}s^{\mu_p}$. If we now assume the scaling behavior for temperature, $T \sim t^{-\beta}$, and substitute it into Eqs. (45) and (47), we find from the condition of consistency the relations (17).

Turn now to more rigorous analysis. As it has been shown above, in the limit of hot granular gas as well as in the cold gas limit, the kinetic kernels of Eq. (8) may be written as:

$$C_{ij} = cT^{\nu_c}\tilde{C}_{ij}; \quad P_{ij} = pT^{\nu_p}\tilde{P}_{ij}, \tag{48}$$

where $c$, $p$, and $\nu_c$, $\nu_p$ are constants and $\tilde{C}_{ij}$ and $\tilde{P}_{ij}$ are the dimensionless homogeneous functions of $i$ and $j$ with the homogeneity exponents $\mu_c$ and $\mu_p$, respectively. We also assume that the evolution of temperature obeys a power law, $T \sim t^{-\beta}$, while the concentrations $n_k$ obey the scaling form,

$$n_k = \frac{1}{t^{2z}}\Phi\left(\frac{k}{t^z}\right). \tag{49}$$

Following the approach of ref.[45], we write the left-hand side of Eq. (9) as:

$$\frac{dn_k}{dt} = -\frac{2z}{t^{2z+1}}\Phi - \frac{1}{t^{2z}}\frac{zk}{t^{z+1}}\Phi' = -\frac{z}{t^{2z+1}}\left(2\Phi + x\frac{d\Phi}{dx}\right), \tag{50}$$

where $x = k/t^z$. This is equal to the right-hand side, which we write, changing from the discrete variables $i$, $j$ to the continuous ones, $C_{ij} \to C(i,j)$ as:

$$\begin{aligned}
\frac{dn_k}{dt} = &\frac{1}{2}cT^{\nu_c}\int_0^k dj\, \tilde{C}(k-j,j)\frac{1}{t^{2z}}\Phi\left(\frac{k-j}{t^z}\right)\frac{1}{t^{2z}}\Phi\left(\frac{j}{t^z}\right) \\
&- cT^{\nu_c}\frac{1}{t^{2z}}\Phi\left(\frac{k}{t^z}\right)\int_0^\infty dj\, \tilde{C}(k,j)\frac{1}{t^{2z}}\Phi\left(\frac{j}{t^z}\right) \\
= &-cT^{\nu_c}t^{\mu_c z - 3z}\left[\int_0^\infty dy\, \tilde{C}(x,y)\Phi(x)\Phi(y) \right. \\
&\left. -\frac{1}{2}\int_0^x dy\, \tilde{C}(x-y,y)\Phi(x-y)\Phi(y)\right].
\end{aligned} \tag{51}$$

where we take into account that $k = xt^z$, $j = yt^z$ and use

$$\tilde{C}(xt^z, yt^z) = (t^z)^{\mu_c}\tilde{C}(x,y) \tag{52}$$

and similar relations that hold true for homogeneous kernels. Hence, Eq. (9) transforms into

$$\begin{aligned}
\frac{z}{t^{2z+1}}\left(2\Phi + x\frac{d\Phi}{dx}\right) = &\, cT^{\nu_c}t^{\mu_c z - 3z}\left(\int_0^\infty dy\, \tilde{C}(x,y)\Phi(x)\Phi(y)\right. \\
&\left. -\frac{1}{2}\int_0^x dy\, \tilde{C}(x-y,y)\Phi(x-y)\Phi(y)\right)
\end{aligned} \tag{53}$$

If we divide both sides of the above equation by $cT^{\nu_c}t^{\mu_c z - 3z}(2\Phi + x\Phi')$, we observe that one side of the equation depends only on $t$ while the other one on $x$ and $y$. This

implies the following condition[45]

$$\frac{z}{c}t^{\nu_c\beta + z(1-\mu_c) - 1} = w = \mathrm{const}, \tag{54}$$

where $w$ is the separation constant. This condition yields Eq. (16) for $\Phi(x)$ and the relation

$$\nu_c\beta + z(1-\mu_c) - 1 = 0. \tag{55}$$

For the scaling solution (49), one can write for $N = \sum_i n_i$:

$$N = \int_0^\infty dk\, \frac{1}{t^{2z}}\Phi\left(\frac{k}{t^z}\right) = Ct^{-z}, \quad C = \int_0^\infty dx\, \Phi(x). \tag{56}$$

Then one can write for Eq. (10):

$$\begin{aligned}
\frac{d}{dt}NT &\sim t^{-z-\beta-1} \\
&= pT^{\nu_p}\int_0^\infty dx \int_0^\infty dy\, \tilde{P}(xt^z, yt^z)\frac{1}{t^{2z}}\Phi(x)\Phi(y) \\
&\sim t^{-\beta\nu_p + z(\mu_p - 2)},
\end{aligned} \tag{57}$$

which gives another scaling relation:

$$-z - \beta - 1 = -\beta\nu_p + z(\mu_p - 2). \tag{58}$$

The scaling relations, Eqs. (55) and (58), correspond to the pairs of scaling relations, Eqs. (18), (20), and (24) for different values of $\nu_c$ and $\nu_p$, that one has for different aggregation regimes.

**Direct simulation Monte Carlo.** To confirm the theoretical results, we perform numerical direct simulation Monte Carlo (DSMC) simulations. DSMC is a numerical technique used to directly solve the Boltzmann equation. It was first elaborated by Bird[59] for the simulation of molecular gases and later generalized and applied to the Enskog equation for dissipative granular gases, see e.g., refs. [60–64]. As there is an extended introductory literature on the application of DSMC to granular gases, e.g. refs. [65,66], here we describe only the details of our approach where we deviate from the standard DSMC.

**Laboratory time for non-aggregating particles.** For reasons which will be clear below, our DSMC algorithm does not rely on the laboratory (real) time but on the number of performed collisions. For the simple case of non-aggregating particles where the total number of particles is preserved, we compute the laboratory time à posteriori from the number of collisions via the collision frequency, $\nu$:

$$\nu = 4\sqrt{\frac{\pi}{2}}\frac{N_p}{V}\sigma^2 v_0 = \left.\frac{C_{\Delta t}}{N_p\Delta t}\right|_{T,N_p = \mathrm{const}}. \tag{59}$$

That is, during a short interval of time, $\Delta t$, where the temperature, $T$, does not change noticeably in a homogeneous system of $N_p$ particles in volume $V$, there occur $C_{\Delta t}$ collisions. Here, we use the notation,

$$v_0 = \sqrt{\frac{2T}{m}}, \quad T = \frac{2E}{3N_p}, \quad E = \sum_i \frac{1}{2}mv_i^2, \tag{60}$$

where $m$ and $\sigma$ are respectively the mass and diameter of the particles. Then an interval of the laboratory time $\Delta t$ may be computed through the number of collisions $C_{\Delta t}$ as:

$$\Delta t = \frac{C_{\Delta t}}{N_p \nu^x}, \tag{61}$$

where we introduce the notation,

$$\nu^x \equiv 16\sqrt{\frac{\pi}{2}}\frac{N_p}{V}\left(\frac{3m}{4\pi\rho}\right)^{2/3}\sqrt{\frac{2T}{m}}\bigg|_{T,N_p = \mathrm{const}}, \tag{62}$$

with $\rho$ being the density of the particles' material. In our simulations we use $\sigma = 1$, $m = 1$, $\rho = 6/\pi$, $T(0) = 1$ and the packing fraction $\eta = (\pi/6)\sigma^3(N/V) = 0.05$, which guarantees the accuracy of DSMC. The initial mean collision time in these units is $\tau_c = \nu^{-1} = 1.48$.

**DSMC simulation of aggregative particles.** When the particles of a dissipative gas agglomerate, obviously, their total number is not preserved. In our simulations, we start with typically $10^8$ monomers and in the course of time larger and larger particles emerge, such that after some time not monomers, but much larger particles dominate. Simultaneously, the total number of particles decreases persistently. After a relatively short period of time it will not be possible to obtain reliable data due to the poor statistics. Therefore, we keep the number of particles in the system approximately constant by regularly expanding the system: we start the simulation with $N_p$ monomers, but when arriving by agglomeration at particle

number $N_p/2$, we duplicate all particles. That is, each particle is replaced by two particles of identical velocity and mass. Effectively, this operation corresponds to doubling the system volume, $V$, which defines a scaling variable

$$F = \frac{V}{V_0}, \tag{63}$$

where $V_0$ is the initial system volume. After the first doubling, $F = 2$, after the second, $F = 4$, etc. Consequently, the data obtained as result of the simulation have to be re-scaled before applying Eqs. (61) and (62):

$$N_p = \frac{N_{MC}}{F}; \ C_{\Delta t} = \frac{C_{\Delta t}^{MC}}{F}. \tag{64}$$

Here, $N_{MC}$ is the number of particles, simulated at the current time corresponding to a certain temperature. Similarly, all other variables with index MC. This way, the number of particles in the simulation is always in the interval $N_p = [N_0/2 + 1, N_0]$, where $N_0$ is the initial number of particles. This expansion of the system size allows to obtain good statistical data independently of the aggregation process.

**Laboratory time for agglomerating particles**. The composition of a granular gas of aggregating particles changes in time. That is, at a given time, the system consists of $N^{(1)}$ monomers, $N^{(2)}$ dimers,…,$N^{(i)}$ $i$-mers. The abundance of monomers can be considered as a subsystem of particles, the same for dimers and, in general, for $i$-mers. For the case of agglomerating particles, we use the fact that in dilute systems, subsystems of particles behave independent of one another. Obviously, since these subsystems coexist, there are many ways to determine the laboratory time from the number of collisions via Eq. (61), where we use the data from subsystem $i$: $N^{(i)}$, $m^{(i)}$, $T^{(i)}$. Consequently, from the simulation results we obtain many different laboratory times, which are, of course, theoretically identical, but not numerically due to the evolution of the system and the finite system size: near the beginning of the simulation, most of the particles are monomers, therefore, the statistics of the monomers delivers the most reliable results while the statistics based on 20-mers is unreliable since the abundance of 20-mers is yet small. For a given particle size, $i$, there is a certain interval of time when the abundance of $i$-mers is the largest subsystem from which we obtain the most reliable data. For earlier time, the statistics is poor since most of the particles are yet smaller than $i$, for later times, the statistics is poor as well since most of the $i$-mers vanished from the system by agglomeration. Consequently, in our simulation we determine the laboratory time from the most reliable species, $I$, consisting of the largest number of particles at the given time. Therefore, we compute

$$F = \frac{V_{MC}}{V_0}; \ N^{(I)} = \frac{N_{MC}^{(I)}}{F}$$
$$C^{(I)} = \frac{C_{MC}^{(I)}}{F}; \ T^{(I)} = \frac{2}{3}\frac{E^{(I)}}{N^{(I)}}$$
$$m^{(I)} = I; \ E^{(I)} = \sum_{i \in I}\frac{1}{2}m^{(I)}v_i^2 \tag{65}$$
$$\nu^x = 16\sqrt{\frac{\pi}{2}}\frac{N^{(I)}}{V_0}\left(\frac{3m^{(I)}}{4\pi\rho}\right)^{2/3}\sqrt{\frac{2T^{(I)}}{m^{(I)}}}.$$

**Data availability**. All relevant data are available from the authors.

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

## Acknowledgements

We thank the German Research Foundation (DFG) for funding through the Cluster of Excellence "Engineering of Advanced Materials" and the Collaborative Research Center SFB814 for financial support. Support from ZISC and IZ-FPS at FAU Erlangen-Nürnberg is gratefully acknowledged.

## Author contributions

N.V.B. and T.P. developed the theoretical results; A.F. and T.P. carried out the numerical simulations. All authors analyzed the data, discussed the results, and prepared the manuscript.

## Additional information

**Competing interests:** The authors declare no competing financial interests.

