## [Peer Review File · Nature Communications]

Reviewers' comments:

Reviewer #1 (Remarks to the Author):

This paper reports on a very interesting and quite counterintuitive phenomenon that concerns the evolution of granular gases that are not driven by external forcing: while the gas as a whole will cool due to dissipative collisions among the particles, the rms velocity fluctuations of particles aggregates that form in the presence of small attractive forces can actually increase, i.e., the effective temperature of the aggregates will actually rise. This phenomenon is here reported for the first time and was missed in the rather extensive prior theoretical and simulation work on freely cooling gases, which focused on purely repulsive particle-particle interactions.

I think this is an excellent paper. However, there is one aspect I am missing, and that is a clear and concise discussion of any observable consequences of the findings. Specifically:

1. Given that small attractive interactions, such as van der Waals or Coulomb forces, are probably present under almost all conditions relevant to situations ranging from powder processing to the formation of protoplanetesimals, one might suppose that aggregate "heating by cooling", as discussed by the authors, should occur in those situations. Do the authors believe this is the case? If so, could the heating be observed experimentally?
2. Would this phenomenon manifest itself in any other way than through the partial temperatures associated with different cluster sizes? Would, for example, the fact that the partial temperatures are different in regime (iii) affect the eventual distribution of observed cluster or aggregate sizes?
3. In Figs. 1 and 2 the heating sets in after roughly 1,000 time steps; what sets this characteristic time scale and how does it translate to the number of actual collisions (what is written in the Methods section seems incomplete in answering this)?

Reviewer #2 (Remarks to the Author):

This paper reports a remarkable phenomenon in aggregating granular gases, that is, in systems in which collisions are either partly inelastic, or else lead to the fusion of the two particles. There, the authors find a phenomenon they call 'heating by cooling' (more on this terminology later), which corresponds to an increase with time of the average kinetic energy accompanying the decrease of the total kinetic energy. The latter indeed always decreases, since it decreases at each collision, but its average can and does increase under appropriate circumstances, since the total number of particles decreases due to the aggregation process.

The paper is very well written, and, as to substance, the effect claimed is convincingly argued for, from a theoretical viewpoint, and it also appears in the numerical simulation the authors provide. So far the good news.

My main objection concerns the expression, 'heating by cooling', which the authors have chosen to characterise the effect. It has several problems. The first is that, if words are used consistently, the expression is simply contradictory: it cannot be claimed that, if heat leaves the system, then the internal energy of the system increases. Of course, in the authors' mind, this expression means that temperature increases as energy decays. But this implies an inconsistent use of the terms heating and cooling, one referring to heat and the other to temperature, thereby confounding efforts to explain the difference between both concepts.

Another, equally serious problem, is that this terminology fails to show the parallelism to a similar well-known phenomenon in equilibrium, namely negative specific heat. I believe a short reference to

the considerable literature on that subject, say Lynden-Bell and Lynden-Bell's works on the subject, or S. Ruffo's, on the presence of negative specific heat in systems with long-range interactions should, at least, be mentioned and compared with what is being described in the paper. Indeed, the long-range systems mentioned above do usually display some form of clumping in the regime of negative specific heat.

As to suggestions concerning a new term, it seems to me that the authors should think this over carefully. One option would, of course, be to replace 'heating by cooling' everywhere by 'negative specific heat'. There may be some issues with this, but it would appear simplest to this referee. Other possible expressions might be 'cooling-induced temperature increase' or anything similar, which would not involve a confusion between temperature and heat.

So far, the changes I am suggesting are important, and I think they really must be made, but they do not affect the paper in its overall structure. In the following, I would suggest a possible rather extensive amendment to the paper, but it is merely a suggestion, which I believe might make the paper easier to read.

In the scaling theory of aggregation, one has 'easy' results, such as those pertaining to typical size, and 'hard' results such as those involving the polydispersity exponent τ or the related monomer decay exponent. For the latter, the use of the formalism developed by Ernst and van Dongen is unavoidable. Such is, however, usually not the case for quantities such as the typical size. Thus one has for the second moment

$$\dot{M}_2 = \sum_{k,l=1}^{\infty} k l K(k,l) c_{k,l}$$

which, using scaling, leads to

$$\dot{s} \propto K(s,s) \propto s^{-\lambda}$$

which immediately gives the well-known result

$$s(t) \propto t^{1/(1-\lambda)}.$$

Now the quantities examined by the authors are all of the type of 'typical size'. I believe the paper would considerably gain if the authors could find a way of formulating their results in a way similar to the one presented above. This would make the paper a great deal easier to read, and the message would be much more efficiently transmitted to the reader. The detailed formalism using reaction rates and the scaling formalism of van Dongen could then still be discussed in the Supplementary Material.

Finally, it seems to me that the use of the two exponents λ_1 and λ_2 is slightly unnecessary, not to say confusing: the degree of homogeneity is given by

$$\lambda = \frac{2\lambda_1 - \lambda_2}{3}$$

and this is indeed the quantity which appears everywhere in the different formulae. Introducing λ as above and systematically using it would greatly simplify understanding of the results presented in the paper.

To summarise, I believe this is a wonderful paper, which should be published once the authors settle the issue of the changes concerning the terminology of 'heating by cooling', as well as make an appropriate discussion of the relationship to negative specific heat. The authors should further decide

as to whether they attempt the simplifications suggested in the second part of this referee report.

Reviewer #3 (Remarks to the Author):

In this paper, the authors analyse the dynamics of a collection of inelastic particles that on collision either (a) rebound inelastically or (b) coalesce if the relative velocity is small enough. The main novel result that is claimed is that while the total energy decreases with time, the mean energy per particle could increase with time. In my opinion, there is nothing novel about this result (see below), neither does it make a conceptual advance that changes the way of thinking in the field. Thus, the paper does not meet the criteria for publication in nature communications, and I recommend rejection.

Let me first give the simple example of ballistic aggregation (only (b) as defined in first paragraph). An aggregate of mass M has velocity $V \sim M^{-1/2}$ by momentum conservation (for example see Ref [26], Carnevale, Pomeau, et al. of paper). Energy of this particle goes as $MV^2 \sim t^0$, and does not decrease with time. Clearly, the total energy decreases, for example as t^{-1} in two dimensions, but the average energy does not decrease. Clearly, this is a well known in the literature and there is nothing surprising or conceptual about the result.

What the authors in the current study have done is to introduce rebound also. Initially, particles cool as Haff's law. As it cools, some particles which have small relative velocities are allowed to coalesce. This is in effect decreasing the number of particles with negligible change in energy (because of selective coagulation) resulting in the mean energy per particle going up. In a way, this is equivalent to removing particles of low energy from the system, resulting in energy per particle going up.

Thus, if I were to restate the result of the paper as "In a cooling gas, if I selectively remove particles of low energy, then the mean energy per particle will increase while the total energy will decrease". This is an obvious statement.

On the other hand, the calculation has merit, and deserves publication in some form in a regular archival journal. The numerics is a direct computer simulation of the Boltzmann equation, and it is no surprise it matches with the calculation. If it did not match, then there would have been some curiosity. The paper can be made more interesting by performing event driven simulations of a system with spatial coordinates, and see if the homogeneous regime shows any of these features.

One minor comment: Eq (6) where a Gaussian is assumed for the velocity distribution. It is quite well established that the velocity distribution in the homogeneous cooling regime is not a Gaussian. For example, it is an exponential in three dimensions [see X. Nie, E. Ben-Naim, and S. Chen, Phys. Rev. Lett. 89, 204301 (2002), T. P. C. van Noije and M. H. Ernst, Granular Matter 1, 57 (1998), Pathak S. N., Jabeen Z., Das D. and Rajesh R., Phys. Rev. Lett., 112 (2014) 038001]

REPLY to reviewer 1:

This paper reports on a very interesting and quite counterintuitive phenomenon that concerns the evolution of granular gases that are not driven by external forcing: while the gas as a whole will cool due to dissipative collisions among the particles, the rms velocity fluctuations of particles aggregates that form in the presence of small attractive forces can actually increase, i.e., the effective temperature of the aggregates will actually rise. This phenomenon is here reported for the first time and was missed in the rather extensive prior theoretical and simulation work on freely cooling gases, which focused on purely repulsive particle-particle interactions.

I think this is an excellent paper.

Our reply: We thank the referee for the kind assessment of our work, underlining the novelty of our findings. We revised the manuscript according to the comments by the referee. Below, please, find our detailed reply. For convenience, changes to the manuscript appear highlighted by blue color.

However, there is one aspect I am missing, and that is a clear and concise discussion of any observable consequences of the findings. Specifically:

1. Given that small attractive interactions, such as van der Waals or Coulomb forces, are probably present under almost all conditions relevant to situations ranging from powder processing to the formation of protoplanetesimals, one might suppose that aggregate “heating by cooling”, as discussed by the authors, should occur in those situations. Do the authors believe this is the case? If so, could the heating be observed experimentally?

Our reply: Thank you for the interesting question. We believe that the effect discussed here could be observed experimentally. To support this, we consider the most generic case of van der Waals forces which manifest as surface adhesion. We use the available experimental data for the restitution coefficient and material parameters (which are, unfortunately, not very abundant) and consider two limiting cases – very hard particles (ceramic grains) and very soft ones (acrylic grains). For both system we find the parameters, which are rather realistic – the particles’ diameters are of the order of 1 mm, while the characteristic velocity is of the order of 0.1 m/s. Based on this we conclude that the regime of temperature increase may be observed in many system, where particles have attractive interactions (see the Supplementary Material for detail).

Experimental investigations of cooling granular gases are not simple – to assure force-free conditions, the action of gravity has to be suppressed. Such experiments have been performed, either under true microgravity conditions aboard the space station [a], in parabolic flights [b,c] or in sounding rockets [d-g] or by means of magnetic levitation [h]:

- [a] Hou et al. “Velocity Distribution of Vibration-driven Granular Gas in Knudsen Regime in Microgravity”, *Microgravity Science and Technology* **20**, 73 (2008)
- [b] Tatsumu et al. “Experimental study on the kinetics of granular gases under microgravity”, *J. Fluid Mech.* **641**, 521 (2009)
- [c] Grasselli et al. “Velocity-dependent restitution coefficient and granular cooling in microgravity”, *Europhys. Lett.* **86**, 60007 (2009)
- [d] Hardt et al. “Three-dimensional (3D) experimental realization and observation of a granular gas in microgravity”, *Adv. Space Res.* **7**, 1901 (2015)
- [e] Hardt et al. “Cooling of 3D granular gases in microgravity experiments”, *Eur. Phys. J. Web Conf.* **140**, 04008 (2017)
- [f] Harth et al. “Granular Gases of Rod-Shaped Grains in Microgravity”, *Phys. Rev. Lett.* **110**, 144102 (2013)
- [g] Siegl et al. “Material Physics Rockets MAPHEUS-3/4: Flights and Developments”, *Proc. 21st ESA Symposium on European Rocket and Balloon Programmes and Related Research* (2013)
- [h] Maaß et al. “Experimental Investigation of the Freely Cooling Granular Gas”, *Phys. Rev. Lett.* **100**, 248001 (2008)

By now, however, the focus of these experiment was to check experimentally the cooling law theoretically predicted by Haff and others [37]. To this end, the authors of the overmentioned papers tried to assure purely repulsive interaction of the particle, that is, to keep attractive forces as small as possible. This might be the reason why the temporary increase of temperature reported in our paper was not mentioned in any of these studies. As it follows from our estimates discussed above and detailed in the

new section of the Supplementary Material (SM), one needs to prepare a system at particular conditions to ensure the emergence of the regime of interest. We believe that using the theoretical results reported in our study, the experimental verification of increasing temperature in a cooling gas will be performed in the nearest future.

We address this point in the amended manuscript. At the end of the first paragraph of the “Conclusion” section we add the sentence: “This surprising effect may be observed for many realistic systems (see SM)”. We also add a new Section “Analysis of the regime of increasing temperature of a cooling gas” to the SM where we discuss in detail the possibility of the experimental observation of the counterintuitive effect of temperature growth.

2. Would this phenomenon manifest itself in any other way than through the partial temperatures associated with different cluster sizes? Would, for example, the fact that the partial temperatures are different in regime (iii) affect the eventual distribution of observed cluster or aggregate sizes?

Our reply: It is not easy to give a mathematically rigorous answer to this question, however one can provide reasonable qualitative arguments based on the scaling properties of the system. In the regime (iii) the kinetic kernels C_{ij} and P_{ij} that determine the evolution of temperature and concentrations are homogeneous functions of clusters sizes, i and j . Hence one can conclude that the resulting aggregate size distribution would be a scaling function $n_k(t) \sim t^{-2z}\Phi(k/t^z)$ (see Eq. (15) of the main text). As it is generally known for granular mixtures (see Refs. [35,36]) and may be also seen from our numerical data, all partial temperatures $T_i(t)$ evolve as a power law with the same exponent, $T_i(t) \sim t^{-\beta} \sim T(t)$. These may have, however, different pre-factors, $T_i(t) = \phi_i T(t)$, where the constants, ϕ_i , weakly depend on i . Hence, under the assumption of equal temperatures, $T_i = T$, one can write $\Phi(k/t^z) = \Phi(kT^{z/\beta})$. Let us now assume that that the substitute $T \rightarrow T_k$ in the argument of the scaling function $\Phi(x)$ would allow to account for different partial temperatures. That is, we apply the substitute:

$$\Phi\left(kT^{z/\beta}\right) \rightarrow \Phi\left(kT_k^{z/\beta}\right).$$

Then, using $T_k = \phi_k T$, we obtain for the cluster size distribution,

$$n_k(t) = \frac{1}{\tilde{t}^{2z}} \Phi(k/\tilde{t}^z) \quad \tilde{t} = t\phi_k^{1/\beta}.$$

Taking into account that ϕ_k weakly depends on k we conclude that the effect of the partial temperature difference will indeed lead to the alteration of the cluster size distribution, but this will not be significant.

3. In Figs. 1 and 2 the heating sets in after roughly 1,000 time steps; what sets this characteristic time scale and how does it translate to the number of actual collisions (what is written in the Methods section seems incomplete in answering this)?

Our reply: In our simulations, we use $\sigma = 1$, $m = 1$, $\rho = 6/\pi$, $T(0) = 1$, and the packing fraction $\eta = (\pi/6)\sigma^3(N/V) = 0.05$. The initial mean collision time in this units is $\tau_c = v^{-1} = 1.48$. Hence the increase of temperature for the parameters corresponding to Figs 1 and 2, starts after approximately 600 collisions.

To estimate the time of the “heating” onset t_h one can use the condition, $(1 - \epsilon^2) < (a/T)^3$, along with Haff’s law for non-aggregating cooling, $T(t) \sim 1/(t/\tau_0)^2$, where $\tau_0 = (1 - \epsilon^2)4\eta/\sigma\sqrt{T(0)}/\pi m$. This yields for the parameters $\epsilon = 0.99$ and $a = 0.1$ of Figs. 1 and 2 the value of $t_h \simeq 500$, which qualitatively agrees with the numerical data. The according estimates for the real systems, discussed above, provide the following estimates for the onset time: $t_h \simeq 17s$ for a gas of 1 mm ceramic particles and $t_h \simeq 72s$ for acrylic particles; this is detailed in SM.

In reply to the referee’s comment, we add to the Methods section of the amended manuscript more details which allow to compute all necessary quantities, e.g. the mean collision time in the exploited time units. Namely we write at the end of the subsection “Computing laboratory time...” , “In our simulations we use $\sigma = 1$, $m = 1$, $\rho = 6/\pi$, $T(0) = 1$ and the packing fraction $\eta = (\pi/6)\sigma^3(N/V) = 0.05$, which guarantees the accuracy of DSMC. The initial mean collision time in this units is $\tau_c = v^{-1} = 1.48$ ”. In the new Section of the SM we add a discussion on the onset time, t_h , for the simulated system and real systems of granular particles.

We wish to thank the reviewer again for the very helpful report which helped to improve the manuscript. We hope that after revision the reviewer can recommend the manuscript for publication.

REPLY to reviewer 2:

This paper reports a remarkable phenomenon in aggregating granular gases, that is, in systems in which collisions are either partly inelastic, or else lead to the fusion of the two particles. There, the authors find a phenomenon they call 'heating by cooling' (more on this terminology later), which corresponds to an increase with time of the average kinetic energy accompanying the decrease of the total kinetic energy. The latter indeed always decreases, since it decreases at each collision, but its average can and does increase under appropriate circumstances, since the total number of particles decreases due to the aggregation process.

The paper is very well written, and, as to substance, the effect claimed is convincingly argued for, from a theoretical viewpoint, and it also appears in the numerical simulation the authors provide. So far the good news.

Our reply: We thank the referee for the kind assessment of our work. We are very grateful to the Referee that he/she finds that *"the paper reports a remarkable phenomenon"* and that this is *"a wonderful paper, which should be published"*. We revised the manuscript according to the comments by the referee. Below, please, find our detailed reply. For convenience, changes to the manuscript appear highlighted by blue color.

My main objection concerns the expression, 'heating by cooling', which the authors have chosen to characterise the effect. It has several problems. The first is that, if words are used consistently, the expression is simply contradictory: it cannot be claimed that, if heat leaves the system, then the internal energy of the system increases. Of course, in the authors' mind, this expression means that temperature increases as energy decays. But this implies an inconsistent use of the terms heating and cooling, one referring to heat and the other to temperature, thereby confounding efforts to explain the difference between both concepts.

Our reply: We understand the Referee's concern. The major idea behind the title "Heating by cooling..." was the intention to stress the stunning behavior of aggregating granular gases. We agree, however, that strictly speaking "heating" implies the increase of heat (energy) in the system. Therefore, following the suggestion by the Referee we decide to alter the title, which now reads "Increasing temperature of cooling granular gases" and complies with conventional terminology. We also substitute everywhere in the text the term "heating" by "increasing temperature".

Another, equally serious problem, is that this terminology fails to show the parallelism to a similar well-known phenomenon in equilibrium, namely negative specific heat. I believe a short reference to the considerable literature on that subject, say Lynden-Bell and Lynden-Bell's works on the subject, or S. Ruffo's, on the presence of negative specific heat in systems with long-range interactions should, at least, be mentioned and compared with what is being described in the paper. Indeed, the long-range systems mentioned above do usually display some form of clumping in the regime of negative specific heat.

Our reply: This is an excellent proposal to put the observed effect into a framework of more general thermodynamical context. Indeed the behavior of an aggregating granular gas may be interpreted as a manifestation of negative specific heat – the energy of the system decreases, while temperature increases. Although it is important to stress this analogy, we believe that the direct application of the term "negative specific heat", used for equilibrium systems, is questionable for our non-equilibrium systems and requires further analysis: For instance, the opposite process of temperature decrease at increasing energy requires the inverse process of collision decomposition of aggregates and hence the extension of the present model. Moreover the problem of thermodynamic non-additivity of systems with a negative heat capacity and its application to our systems needs a more detailed analysis. This is a very interesting problem which we leave for future investigations.

In the revised text we mention the analogy in the behavior of our system and systems with negative heat capacity. Namely, in the paragraph located just above the section "Cold-gas limit" we write: "In this regime the increase of the gas temperature occurs during its cooling, hence it behaves like a system with a negative specific heat (see e.g. Refs. [46,47] and SM)". We also add the following sentence at the end of the first paragraph of the right column of page 5: "Interestingly, the increase of temperature with decaying energy corresponds to a negative specific heat for equilibrium systems [46,47]; the application of this concept to non-equilibrium systems requires however further analysis (see SM)". In the Supplementary Material we present a discussion of the analogy between our systems and these with a negative heat capacity as well as the possibility of application of this concept to the systems of interest.

As to suggestions concerning a new term, it seems to me that the authors should think this over carefully. One option would, of course, be to replace 'heating by cooling' everywhere by 'negative specific heat'. There may be some issues with this, but it would appear simplest to this referee. Other possible expressions might be 'cooling-induced temperature increase' or anything similar, which would not involve a confusion between temperature and heat.

Our reply: We agree. See our response to the previous comment.

So far, the changes I am suggesting are important, and I think they really must be made, but they do not affect the paper in its overall structure. In the following, I would suggest a possible rather extensive amendment to the paper, but it is merely a suggestion, which I believe might make the paper easier to read.

In the scaling theory of aggregation, one has 'easy' results, such as those pertaining to typical size, and 'hard' results such as those involving the polydispersity exponent τ or the related monomer decay exponent. For the latter, the use of the formalism developed by Ernst and van Dongen is unavoidable. Such is, however, usually not the case for quantities such as the typical size. Thus one has for the second moment

$$\dot{M}_2 = \sum_{k,l=1}^{\infty} k l K(k,l) c_k c_l$$

which, using scaling, leads to

$$\dot{s} \propto K(s,s) \propto s^\lambda$$

which immediately gives the well-known result

$$s(t) \propto t^{1/(1-\lambda)}.$$

Now the quantities examined by the authors are all of the type of 'typical size'. I believe the paper would considerably gain if the authors could find a way of formulating their results in a way similar to the one presented above. This would make the paper a great deal easier to read, and the message would be much more efficiently transmitted to the reader. The detailed formalism using reaction rates and the scaling formalism of van Dongen could then still be discussed in the Supplementary Material.

Our reply: We agree that it would be better to present such simplified analysis, leaving more rigorous derivation to the Supplementary Material. We tried to do this (see the added new subsection in the SM), but found that: (i) it requires lengthy explanations of additional quantities and rather unrigorous arguments and (ii) it does not allow to consider uniformly all the addressed cases. This makes the narration somewhat "ragged". Therefore we decide to leave the previous way of material presentation.

Nevertheless, we find the suggestion of the Referee appealing and add to the Supplementary Material a new subsection "Qualitative approach", where we present the derivation of the scaling relations exactly in the way proposed by the Referee. We believe that this will help a reader to understand better the scaling method.

Finally, it seems to me that the use of the two exponents λ_1 and λ_2 is slightly unnecessary, not to say confusing: the degree of homogeneity is given by

$$\lambda = \frac{2\lambda_1 - \lambda_2}{3}$$

and this is indeed the quantity which appears everywhere in the different formulae. Introducing λ as above and systematically using it would greatly simplify understanding of the results presented in the paper.

Our reply: Thank you for this suggestion which simplifies the notation. Indeed, $(2\lambda_2 - \lambda_1)/3 = \Lambda/3$ is the degree of homogeneity of the energy aggregation barrier W_{ij} defined by Eq. (2) of the main text. Moreover, the condition $2\lambda_1 - \lambda_2 > 0$ for the possibility of observation of the surprising increase of temperature in a cooling gas has a clear physical meaning: $\Lambda > 0$ implies that the attractive energy barrier increases with the aggregates' size.

Taking into account the physical meaning of Λ , we use Λ everywhere in the revised text instead of $2\lambda_1 - \lambda_2$ (for the notation simplicity we prefer to use Λ instead of $\lambda = \Lambda/3$ as the Referee suggests). Still, however in the definition of the energy barrier W_{ij} in Eq. (2) we keep two exponents λ_1 and λ_2 , since different combinations of these parameters describe different physical mechanisms.

In the revised text we also mentioned that the condition of temperature increase $\Lambda > 0$ has a clear physical meaning – the energy of the aggregation barrier should increase with the aggregates' size. Namely, just after Eq.(21) we write: "...If $\Lambda > 0$, which corresponds to the increase of the attractive barrier with aggregates' size, the exponent β is negative"...

To summarise, I believe this is a wonderful paper, which should be published once the authors settle the issue of the changes concerning the terminology of 'heating by cooling', as well as make an appropriate discussion of the relationship to negative specific heat. The authors should further decide as to whether they attempt the simplifications suggested in the second part of this referee report.

Our reply: We thank the referee for the careful report which helped to improve the manuscript. We hope that, after revision, the referee can recommend publication.

REPLY to reviewer 3:

In this paper, the authors analyse the dynamics of a collection of inelastic particles that on collision either (a) rebound inelastically or (b) coalesce if the relative velocity is small enough... Thus, the paper does not meet the criteria for publication in nature communications, and I recommend rejection.

Our reply: We are glad to know that the Referee finds that “*the calculation has merit*”, however, we do not agree with his/her main criticism. In more detail:

The main novel result that is claimed that while the total energy decreases with time, the mean energy per particle could increase with time. In my opinion, there is nothing novel about this result (see below), neither does it make a conceptual advance that changes the way of thinking in the field.

Our reply: We do not agree, since the result is novel, it has not been reported before. The example of such “known result”, given by the Referee is not relevant (see below). Moreover, we believe that our findings, where we reveal a new counterintuitive evolution regime and formulate its condition, may be important for numerous applications, including planetary science.

Let me first give the simple example of ballistic aggregation (only (b) as defined in first paragraph). An aggregate of mass M has velocity $V \sim M^{-1/2}$ by momentum conservation (for example see Ref [26], Carnevale, Pomeau, et al. of paper). Energy of this particle goes as $MV^2 \sim t^0$, and does not decrease with time. Clearly, the total energy decreases, for example as t^{-1} in two dimensions, but the average energy does not decrease. Clearly, this is a well known in the literature and there is nothing surprising or conceptual about the result.

Our reply: We are very surprised by this comment of the Referee and find it as not relevant. Indeed, our main finding is the *increasing* temperature of an aggregating granular gas. In contrast, the example given by the Referee refers to the case of *constant* average energy (temperature). We believe that there is a principal difference between the two evolution regimes: The one with increasing temperature, which is only possible when bouncing collisions are included (as in our study), and the regime of constant temperature, when only aggregative collisions (as in previous studies) may occur. Let us reiterate that our result of increasing temperature is new and not known in the literature, despite of about three decades of intensive research in the field of granular gases. We refer also to the two other Referees who find our result new, conceptual and surprising.

What the authors in the current study have done is to introduce rebound also. Initially, particles cool as Haffs law. As it cools, some particles which have small relative velocities are allowed to coalesce. This is in effect decreasing the number of particles with negligible change in energy (because of selective coagulation) resulting in the mean energy per particle going up. In a way, this is equivalent to removing particles of low energy from the system, resulting in energy per particle going up.

Our reply: We agree with the first part of the Referee’s statement but strongly disagree with the second part. It is true, that we consider systems with a general (realistic) model of particles’ interactions, which describes not only aggregative, but also bouncing collisions. We also agree that the mechanism of temperature increase is related to the coalescence of particles with small relative velocities. But we disagree with the statement that “this is equivalent to removing particles of low energy from the system”. We *do not* remove particles of low energy from the system. Just the opposite – the coalescing particles may have both very large energy, only the relative tangential velocity matters.

Thus, if I were to restate the result of the paper as “In a cooling gas, if I selectively remove particles of low energy, then the mean energy per particle will increase while the total energy will decrease”. This is an obvious statement.

Our reply: We disagree and believe that the Referee incorrectly understood the mechanism of temperature increase. Our result *may not* be restated in the form proposed by the Referee. As we already noted above, we do not selectively remove particles of low energy, neither literally, nor in a figurative sense. The aggregating particles can both have high energy.

In our article we perform a *detailed analysis of the physical aggregation mechanism* and show that not all mechanisms can lead to the temperature growth. The necessary condition for the emergence of this aggregating regime is the positive homogeneity degree of the aggregative barrier, which physically means that the energy of the aggregative barrier increases with the aggregates’ size. Such important result that relates the evolution regimes with the properties of attractive interactions has not been reported before. The predicted theoretically regimes have been then confirmed in the numerical simulations.

On the other hand, the calculation has merit, and deserves publication in some form in a regular archival journal. The numerics is a direct computer simulation of the Boltzmann equation, and it is no surprise it matches with the calculation. If it did not match,

then there would have been some curiosity. The paper can be made more interesting by performing event driven simulations of a system with spatial coordinates, and see if the homogeneous regime shows any of these features.

Our reply: We agree with the Referee that the event driven simulations (EDS) would be indeed interesting to perform. The granular gases however are rarified systems, for which the Direct Simulation Monte Carlo (DSMC) provide very accurate results, see e.g. Refs.[48, 52, 55] of the article. The main advantage of the (DSMC), exploited in our study is the possibility of applying the computation trick of the system doubling (see the Section “Methods”). This is necessary to collect an appropriate statistics for partial temperatures of aggregates. We do not see how to apply this (or a similar) technique in EDS.

One minor comment: Eq (6) where a Gaussian is assumed for the velocity distribution. It is quite well established that the velocity distribution in the homogeneous cooling regime is not a Gaussian. For example, it is an exponential in three dimensions [see X. Nie, E. Ben- Naim, and S. Chen, *Phys. Rev. Lett.* 89, 204301 (2002), T. P. C. van Noije and M. H. Ernst, *Granular Matter* 1, 57 (1998), Pathak S. N., Jabeen Z., Das D. and Rajesh R., *Phys. Rev. Lett.*, 112 (2014) 038001]

Our reply: Certainly we know that the velocity distribution in granular gases is not Gaussian and we stated this in the introduction of the initial version of the paper (middle of the left column of page 1). There we also cited some relevant papers, Refs.[10-13], including the one, mentioned by the Referee. It has been however indicated in Refs.[40, 41] and in our previous studies, Ref.[27] and Ref. [10] of the Supplemental Material, that the deviations from the Gaussian do not noticeably affect the cooling coefficient and other kinetic coefficients computed in the article. Hence the Maxwellian velocity distribution provides an acceptable accuracy and may be safely used in the according calculations.

Following the suggestion of the Referee we add references to the papers by Nie, et al. and Pathak, et al. and also mention explicitly in the Supplementary Material, that the use of the Maxwellian distribution provides an acceptable accuracy for the kinetic coefficients.

We thank the referee for the careful report which helped us to improve the manuscript. We hope, the reviewer can recommend the revised manuscript for publication.

REVIEWERS' COMMENTS:

Reviewer #1 (Remarks to the Author):

I have read the replies by the authors to the reviewers' comments and believe that they very adequately addressed any concerns. The revised manuscript has been improved considerably. This paper breaks new ground and I recommend publication.

Reviewer #2 (Remarks to the Author):

In my opinion the authors have answered the various points raised in my report in a satisfactory manner. I am not sufficiently knowledgeable in the subject to make appropriate comments on the points raised by the other referees.

Reviewer #3 (Remarks to the Author):

The authors have responded to all the comments. However, none of their arguments convince me otherwise that the results are novel enough for publication in Nature Communications. Their main argument is that the other two referees find it interesting, and that the simple well-known example that I gave is one of energy/particle not decreasing with time, while they show that energy/particle not only not decrease with time but may increase. As I wrote in my previous report there is no principle constraining the increase or decrease of energy/particle when the number of particles are changing with time. In fact, within the ballistic aggregation model, it is quite trivial to modify the collision rules to preferably collide particles with perpendicular velocities.